# Dynamic Phase and Polarization Modulation Using Two-Beam Parallel Coding for Optical Storage in Transparent Materials

**DOI:** 10.3390/s22229010

**Published:** 2022-11-21

**Authors:** Jintao Hong, Jin Li, Daping Chu

**Affiliations:** 1Centre for Photonic Devices and Sensors, University of Cambridge, 9 JJ Thomson Avenue, Cambridge CB3 0FA, UK; 2School of Instrumentation and Optoelectronic Engineering, Beihang University, Beijing 100191, China

**Keywords:** holography, phase modulations, polarization modulations, SLM

## Abstract

In this paper, we propose and experimentally demonstrate a parallel coding and two-beam combining approach for the simultaneous implementation of dynamically generating holographic patterns at their arbitrary linear polarization states. Two orthogonal input beams are parallelly and independently encoded with the same target image information but there is different amplitude information by using two-phase computer-generated holograms (CGH) on two Liquid-Crystal-on-Silicon-Spatial-Light Modulators (LCOS SLMs). Two modulated beams are then considered as two polarization components and are spatially superposed to form the target polarization state. The final linear vector beam is created by the spatial superposition of the two base beams, capable of controlling the vector angle through the phase depth of the phase-only CGHs. Meanwhile, the combined holographic patterns can be freely encoded by the holograms of two vector components. Thus, this allows us to tailor the optical fields endowed with arbitrary holographic patterns and the linear polarization states at the same time. This method provides a more promising approach for laser data writing generation systems in the next-generation optical data storage technology in transparent materials.

## 1. Introduction

There has been a recent breakthrough in 5D optical storage in transparent materials and it paves a promising way to unlimited lifetime data storage [1,2]. A permanent nanostructure (usually referred to as voxel) with sub-wavelength diameters can be formed in the silica and used to carry data bits. Laser beam direct writing technology is a necessary step to the successful implementation of voxel writing in optical storage [3,4,5,6]. High-quality laser writing beam generation mainly relies on an efficient optical field modulation approach to encode both intensity information and polarization states of light. This is because the combination of these two parameters can increase the data recording capability of each data voxel by two orders of magnitude. For example, a femtosecond laser writing light field can provide four different linear polarization angles: 0°, 45°, 90°, and 135°. Combining two different pulse durations in writing, we can acquire eight different coding states for data storage. This means that when each coding state can be expressed by a three-digit binary code (i.e., 000, 001, 010…, 111), a long series of binary codes can be encoded into a 2D array of voxels on a data page in a glass block. Accordingly, when the writing beams have more available linear polarization states, the data bits carried by one voxel will increase since the total number of coding-states combinations increases. Therefore, the polarization state is an important property of the writing light field in an optical storage task. An efficient and precise polarization modulation method is very helpful for laser beam direct writing generation in optical storage in transparent materials.

Several works have been reported on achieving light polarization modulations. Based on the current investigation states, we have grouped them into two categories as follows. The first category of methods is polarization modulations for laser writing beam generations. The existing polarization modulation methods for laser writing beam generations can be further summarized into three methods. The first method is based on two SLMs methods. Allegre et al. adopted a Z shape structure by using two-phase-only SLMs in conjunction with a pair of wave plates to control the wavefront and polarization of a laser beam [7,8]. In [9], Hasegawa et al. developed a complex polarization modulation method based on a pair of SLMs, an HWP, and a QWP, to create a holographic vector wave femtosecond laser processing system. The second method is based on single SLM methods. Allegre et al. constructed a polarization beam generation method using a single SLM and a λ/4 wave plate [10]. In [11], Lam et al. developed a beam modulator composed of a transmissive SLM and a quarter-wave plate, which is capable of generating continuously rotating polarization by combing cross-polarizations. In [12], Ono et al. utilized a single SLM, two QWPs, and four HWPs to generate a vector hologram beam, where the SLM controlled the polarization distribution in the cross-section of the two writing beams. An efficient polarization method based on a single SLM was developed by a University of Southampton group [13,14,15,16,17]. They utilized an LCOS SLM and a rotation-free half-wave plate matrix (HPM) to obtain the data pattern generation and the writing beam’s polarization state control. The third method is a simple method, only utilizing a rotatable half-wave plate (HWP). In voxel writing, only linearly polarized light is used to generate the writing beam, thus, a rotatable half-wave plate (HWP) is a direct way to change the angle of a linearly polarized light. However, utilizing a rotatable HWP in voxel writing beam generation has a challenging problem: the significant decrease in writing speed due to the rotation of the HWP.

Apart from the laser writing beam applications, some polarization light generation methods have also been reported. In [18], a polarization control method using a single SLM through three cascaded interactions was investigated to avoid the large light loss. In [19], a parallel aligned nematic SLM was utilized to generate complete arbitrary spatially variant polarization modulation. In [20], the authors utilized a single digital hologram presented on an SLM to implement the simultaneous generation of many vector beams. They demonstrated the simultaneous generation of 16 vector vortex beams with various polarization distributions and spatial shapes on a single SLM. In [21], a reflective twisted nematic SLM, in combination with a beam splitter and a quarter-wave plate, was utilized to implement a polarization rotator. Their experimental results indicated that a curve of the behavior of this polarization rotator proves a modulation up to 177°, possibly suitable for the engineering of cylindrical vector fields. In [22], the system can be regarded as the combination of two variable retarders with tunable retardance, with a relative orientation of 45° among them. Although the system introduces some important losses, mainly due to the four passages through the beam splitters, it has been experimentally validated to act as a polarization generator covering the complete Poincar’e sphere. In [23], a spatial polarization modulation method was reported through a phase-only SLM utilized in a configuration between two quarter-wave plates. Actually, these methods mainly focused on the spatial polarization generation. In [24], a prism and grating-free setup built around a single phase-only spatial-light-modulator were constructed to implement full control of the spatial intensity, phase, and polarization distributions. In [25], Chen et al. utilized an SLM and two half-wave plates or quarter-wave plates to implement the amplitude, phase, and polarization control. This method mainly depends on a macro-pixel encoding method to achieve polarization control. In [26], Pascuala et al. presented a method of generation of arbitrarily polarized vector beam modes by employing two SLMs. Each SLM displayed a different phase-only mask, with each one encoding a different pattern onto two orthogonal linear polarization components of the input beam. Although current polarization modulation methods can control the polarization, they are not suitable for the 5D optical storage application because most of them still cannot achieve arbitrary linear polarization state generation and phase coding at the same time.

Other polarization control works have been reported in other application cases. In [27], a polarization control method was reported for focus-shaping in high-NA microscopy. In scattering imaging [28], a full polarization wavefront correction system was developed to shape the scattered light wavefront in two orthogonal polarizations with a single LC-SLM. In [29], a digital optical phase conjugation was used to achieve polarization modulation for optical focusing scattering media. In the digital laser [30], a single intra-cavity SLM was utilized to implement a digital laser for on-demand modes with polarization control. This method can digitally control and switch basing modes with desired linear polarization at video rates. In [31], a polarization-dependent phase modulation using a single SLM was developed to implement the rotating point spread functions in super-localization. However, those methods were usually developed for the generation of spatially verified polarization distribution, for example, vortex polarization light field for optical tweezers. However, in the application of optical data storage in glass, each voxel writing only requires one uniform linear polarization state of light. This means that the required writing light field could usually be a discrete dot array and each dot with its own linear polarization state instead of an integral beam in previous polarization modulation applications. Therefore, a novel linearly polarized light controlling method is needed for voxel writing.

From the existing methods, current polarization approaches still have limited modulation capability, specifically, in the simultaneous implementation of arbitrary linear polarization and phase patterns. Alternately, in this paper, we hope to develop new polarization modulation approaches with the simultaneous generation of the arbitrary phase and polarization modulations for the future applications of laser beam generations. We propose a dynamic coding of two parallel beams, which is capable of the simultaneous implementation of dynamically generating holographic patterns and arbitrary linear polarization states. This method can provide a significant increase in information density stored in each voxel. More importantly, due to the benefits of using computer-generated holograms (CGHs) on an LCOS SLM, the arbitrary dots pattern can be created. With these two major advantages, our method makes it possible to write all voxels with the same polarization direction on a data page at the same time only using LCOS SLMs. For example, when there are eight different polarization angles available for a voxel, it only requires eight writing cycles to finish the whole data page. This will lead to less movement of glass blocks in the data writing process. It means that a significant decrease in the mechanical delay is caused by glass block platform movement. Since the mechanical delay takes up a major part of the current writing process time duration, our method can largely increase the data writing speed. Furthermore, as no external active optical element is required, our method can be a potential plug-in technology in laser processing.

The remainder of the paper is organized as follows. In Section 2, we describe the proposed polarization and phase modulation methods using two parallel beam coding. In Section 3, the arbitrary holographic vectorial patterns are experimentally generated to verify the effectiveness of the method. In Section 4, the comparison of the proposed method and existing method for generating data writing light field is discussed. We conclude this work in Section 5.

## 2. Proposed Method

### 2.1. Principle

The proposed phase and polarization modulation method can simultaneously modulate the phase and polarization state of a light field. In this method, phase modulation means the holographic pattern information encoding of the input light field. This modulation is achieved by using a CGH on a phase-only LCOS SLM. The polarization modulation here means the control of the polarization state of the output holographic pattern. The importance of this method is that the two targets, desired holographic pattern and desired polarization state, can be achieved at the same time. Both phase and polarization modulations are simultaneously completed by using LCOS SLMs and there are no external special devices needed.

When considering the modulation of the linear polarization state of a beam, if we can independently control the amplitude ratio and phase shift between the vertical and horizontal polarization components of the light, we can achieve any polarization state that we need. As shown in Figure 1, two orthogonally polarized beams are independently encoded with holographic patterns and amplitude information by using two LCOS SLMs. The two modulated beams are then combined into one output light field. The CGH on an LCOS SLM can carry both the holographic image and amplitude information. The amplitude of the holographic vectorial light field is modulated by changing the phase depth of the CGH. By controlling the amplitude ratio of two basis beams, the polarization direction of the combined beam can be altered. This will be introduced in detail later in this section. To acquire the output holographic vectorial pattern, the combining of two modulated beams plays an important role in this method. The principle and combining method are also explored and covered in the following sections.

### 2.2. The Encoding Principle of the Proposed Method

Figure 2 shows the coding principle of the proposed phase and polarization modulation method. First, two coherent laser beams are inputted into two separate optical vector channels (horizontal and vertical channels). The two independent channels are used to produce two basis holographic vector components with an orthogonal polarization direction for their later combination. The phase-only LCOS SLMs are sensitive to the polarization state of the input light due to the operating principle of liquid crystal modules [32,33,34,35,36]. This means that the polarization state of input light needs to be parallel to the alignment direction of liquid crystal molecules for phase-only modulation. In this method, we utilize phase modulation to encode holographic image information on the beams. Therefore, the polarization state of input beams in two optical channels is first changed into parallel with the liquid crystal working direction to ensure the high-quality holographic information is encoded on two channels. In the practical design, two LCOS SLMs are parallelly placed for the symmetrical optical path; thus, both optical channels provide phase modulation at a horizontal polarization state.

After two vector beams are modulated horizontally, there needs to be a process to convert one of them into vertically polarized beams for polarization modulation. In the horizontal channel, the hologram on the LCOS SLM directly encodes the horizontal polarization input beam to produce the horizontal basis vector. In the vertical channel, the vertical vector beam is generated by using a half-wave plate (HWP) after passing the LCOS SLM. The modulated horizontal vector beam passes through an HWP at 45°. The HWP is then used to rotate the horizontal polarization state into the vertical state. The horizontal and vertical vector components with the given holographic image and amplitude information are now formed. Finally, two orthogonal vector beams are combined into one beam with a single linear polarization state using a beam combiner.

By applying this proposed method, we can independently modulate the phase and polarization information of the light field. This can fundamentally increase the whole data page writing speed because the glass sample movement is avoided. Moreover, the changing of data pattern and polarization direction can be simply performed by changing the CGH on the LCOS SLM. Here is the theory of this proposed method based on the Jones matrix. Let two computer-generated holograms, denoted by $G_{V}\left(x,y\right)=e^{i\varphi_{V}\left(x,y\right)}$ and $G_{H}\left(x,y\right)=e^{i\varphi_{H}\left(x,y\right)}$, be presented on two SLMs in the vertical and horizontal vector channel, respectively. To consider a simple case, we analyze the zero and first order of the output light field after modulation. When the incident light in the vertical channel is modulated by the LCOS SLM and its vector direction is rotated by 90° after the HWP, the Jones vector of the observed light field here can be expressed as:(1)$$E_{V}=\left[\begin{array}{c} 1\\ 0 \end{array}\right]A_{1}^{V}\cdot e^{i\varphi_{V}\left(x,y\right)}$$
where $A_{1}^{V}$ is the first-order intensity of the diffraction light at the vertical optical channel. At the same time, the horizontal channel is modulated by another LCOS SLM. The Jones vector of the observed light field here can be expressed as:(2)$$E_{H}=\left[\begin{array}{c} 0\\ 1 \end{array}\right]A_{2}^{H}\cdot e^{i\varphi_{H}\left(x,y\right)}$$

The amplitude and holographic target image information has been stored in the form of phase in the beam after two independent phase modulations. To acquire two reconstructed light fields with target intensity and an image for further combination, two converging lenses are used to move the diffraction of light from the infinite far field to the focal plane of a lens.

When the two optical vector channels are provided with the same target image information, i.e., $e^{i\varphi_{V}\left(x,y\right)}=e^{i\varphi_{H}\left(x,y\right)}=e^{i\varphi \left(x,y\right)}$, two orthogonal linearly polarized light fields can be superposed into one beam with a final linear polarization state. By combining the two orthogonal polarization components, we can obtain the final light field with the target image and specific polarization angle programmed by the CGHs on an LCOS SLM. A beam combiner is used to complete the spatial superposition of two orthogonal polarization beams. The two base vectors are combined into an optical vector as:(3)$$E=E_{H}+E_{V}=\left[\begin{array}{c} 1\\ 0 \end{array}\right]A_{1}^{V}\cdot e^{i\varphi_{V}\left(x,y\right)}+\left[\begin{array}{c} 0\\ 1 \end{array}\right]A_{2}^{H}\cdot e^{i\varphi_{H}\left(x,y\right)}$$
(4)$$E=\left[\begin{array}{c} A_{1}^{V}\\ A_{2}^{H} \end{array}\right]\cdot e^{i\varphi \left(x,y\right)}$$

The first-order intensity of diffraction has a relationship and can be controlled by the phase depth of the hologram. Based on Equation (3), the polarization angle of the final optical vector of first-order light depends on the amplitude ratio of the vertical and horizontal polarization components of first-order light. The same principle is suitable for zero-order light. We can effectively control the amplitude distribution (*AV1*, *AH1*) of two orthogonal polarization components of light to generate an arbitrary linear polarization state. Based on Equations (3) and (4), the simultaneous implementation of the polarization modulation and holographic image can be achieved. Therefore, the proposed method can provide the simultaneous phase and polarization modulation of light, and in the application of writing beam generation in optical data storage, this method can dynamically generate a target foci array with the wanted linear polarization state.

### 2.3. The Control of Polarization Angle of Output Beam

To control the polarization angle of the output vector beam, we need to first understand the parameter that determines the polarization state of light. When we split a polarization state into two orthogonal polarization components in a coordinate, there are two parameters that can be used to precisely define a specific polarization state of light: the amplitude ratio and phase shift between two polarization components. As shown by examples in Figure 3, when both polarization components have the same amplitude, the phase shift can determine the final polarization state. When the phase shift is 0, the amplitude ratio can determine the angle of linear polarization. The extreme case is when one of the polarization components is 0; the final polarization state can be either linear vertical or horizontal polarization. Therefore, we can effectively manipulate the output linear polarization angle by altering the amplitude ratio and maintaining the phase shift at 0 at the same time.

As shown in Equation (4), we acquire the Jones vector of the output vector beam. To examine the polarization state of the output beam, we can rewrite the Jones vector as:(5)$$E=\left[\begin{array}{c} A_{1}^{V}\cdot e^{i\varphi \left(x,y\right)}\\ A_{2}^{H}\cdot e^{i\varphi \left(x,y\right)} \end{array}\right]$$

We can see that due to the use of the same holographic target image information on both horizontal and vertical polarization components, the phase distribution encoded on both polarization components can be identical. The phase distribution encoded by a CGH can lead to the identical target image reconstruction process (diffraction of light) in the far field. Therefore, in the proposed method, the phase shift between two components remains 0 after the combination of two basis beams. This fact provides us with one of the conditions for a linear polarization state: the phase shift between two polarization components remains at 0.

To control the angle of the linear polarization state, we now need to control the amplitude of two basis beams: AV1 and AH1. The holographic imaging is based on the diffraction of light in the far field or at the focal plane of a lens. This means that the property of the two basis beams follows the characteristics of the diffraction of light. Therefore, in the proposed method, the accurate control of the diffraction efficiency of a computer-generated hologram is the key to the intensity control of the two basis vector beams.

The diffraction efficiency is the key parameter for the successful use of LCOS SLMs. Diffraction efficiency can be altered by different sources in the optical system. First, the pixelated structure of an LCOS SLM can reduce the diffraction efficiency in each order [37]. However, due to the recent improvement of the fill factor of the pixel electrode (ratio between pixel area size and pixel pitch), the diffraction efficiency reduction caused by the LCOS devices has become less [38,39]. When the pixel size is further reduced, other sources of diffraction efficiency decrease in LCOS SLMs are caused by the driving of liquid crystal molecules. For example, the fringing-field effect of the electrical field and phase flicker causes a phase fluctuation effect [40,41]. Another source of diffraction efficiency altering is the complex modulation of an LCOS SLM. Any deviation from a complete phase-only modulation with 2π modulation depth can result in diffraction efficiency degradation [42]. Thus, the controllable efficiency reduction caused by a limited phase modulation depth can be used to develop a technique to encode amplitude information onto a phase grating (computer-generated hologram) on an LCOS SLM.

In a computer-generated hologram on an LCOS SLM, the different phase depth used in hologram generation can lead to different diffraction efficiency in each diffraction order. The phase depth of a hologram means the maximum grey level the hologram used. As shown in Figure 4, it is a part of the CGH phase profile of a target image generated under three different phase depths.

We can see that for three holograms of the same target image, the blazed gratings-like phase profile has the same period pattern under three different phase depths (see Figure 5). The difference between holograms is the maximum grey level and the related phase gradient at each periodic pattern. Because the amplitude of the first-order diffraction can be controlled by using phase depth, we can apply this technique to both horizontal and vertical optical channels to independently control their intensity. The polarization angle of the final output beam can now be actively controlled by combining two intensity-modulated basis beams.

To make the proposed method feasible in practical applications, we first need to discover the relationship between first-order intensity and the corresponding phase depth of a computer-generated hologram. We can start by simulating the diffraction behavior of a blazed grating. This is because the CGH of an image can be seen as a combination of many blazed gratings across the whole hologram plane.

To study the diffraction efficiency of the blazed gratings, the phase profile of a blazed grating needs to be determined. Due to the finite pixel resolution of the LCOS SLM, a blazed grating on LCOS devices is a stepped grating as shown in Figure 6. As can be seen in the phase profile, M (0 < *M* < 1) indicates the maximum of the phase values of a grating, and the maximum phase depth is 2π according to the default configuration of most phase-only LCOS SLMs.

The intensity of each diffraction order can be calculated from the Fourier series coefficients of the phase grating. It is a straightforward calculation to derive the Fourier coefficients of the stepped phase grating [43].
(6)$$C_{\mu }=e^{i\frac{\pi \mu }{N}}\frac{e^{-i\pi \left(M-\mu \right)}}{e^{-i\frac{\pi }{N}\left(M-\mu \right)}}\cdot \frac{sin\left(\frac{\pi \mu }{N}\right)}{\pi \mu }\cdot \frac{sin\left(\pi \left(M-\mu \right)\right)}{sin\left(\frac{\pi }{N}\left(M-\mu \right)\right)}$$
where *µ* is the diffraction order number.

Therefore, by analyzing the Fourier coefficients of the grating, the relative efficiency of each diffraction order can be expressed as:(7)$$\alpha_{\mu \left(M,N\right)}={\left|C_{\mu }\right|}^{2}={\left[\frac{sin\left(\frac{\pi \mu }{N}\right)}{\pi \mu }\right]}^{2}\cdot {\left[\frac{sin\left(\pi \left(M-\mu \right)\right)}{sin\left(\frac{\pi }{N}\left(M-\mu \right)\right)}\right]}^{2}$$
When we consider a case for a continuous phase grating where the step number is infinite, the resulting diffraction efficiency can be expressed as:(8)$$\alpha_{\mu \left(M,N\to \infty \right)}={\left|C_{\mu }\right|}^{2}={\left[\frac{sin\left(\pi \left(M-\mu \right)\right)}{\pi \left(M-\mu \right)}\right]}^{2}=sinc^{2}\left(M-\mu \right)$$
where sinc(x)=sin(πx)/(πx). When M=0, the energy in the first-order diffraction is 0. When the phase depth changes in the range 0<M<1, the energy splits on the first order vary. When M=1, the energy is entirely in the first order. This is the fundamental situation that was utilized to encode amplitude information in phase modulation using computer-generated holograms. The amplitude of the first-order diffraction of the blazed phase gratings has a direct positive correlation with the maximum grey level used on the hologram.

Based on Equations (6)–(8), the intensity changes of diffracted orders with respect to the depth control parameter *M* are shown in Figure 7. We can observe that the intensity of the first order increases when the phase depth is increased from 0 to 2π. When *M* = 1, the corresponding grey level of the SLM is 255, while the corresponding grey level of the SLM is 0 when *M* = 0. Therefore, the two SLMs can utilize this property to independently control the intensity of two components when they are sequentially modulated. Since the amplitude ratio of the two-beam vector basis is controlled, we can generate an arbitrary linear polarization state of the final optical vector beam.

## 3. Experiments

### 3.1. Experimental Setup

To verify the feasibility of the proposed method, we experimentally demonstrate the capability of generating a vector beam with the encoded holographic information. Figure 8 shows the experimental setup of the concept-of-proof system. First, a coherent laser beam is collimated and expanded by two lenses (Lens1 and Lens2). A linear polarizer (LP) then changes the expanded beam into a horizontal linearly polarized beam. Next, a beam splitter (BS) is used to split the horizontal polarization beam into two sub-beams, forming the input of the two base vectors. The two horizontally polarized beams are then encoded with the holographic information that is presented on the two LCOS SLMs. The Fourier holograms here are calculated based on the target image pattern. A half-wave plate (HWP) is then applied to rotate one of the modulated horizontal polarization beams into a vertically polarized beam. Finally, a beam combiner is employed to superpose the two base vectors into a designed linear vector beam. The holographic pattern is imaged via 4f-relay optics and captured by a CMOS sensor.

The laser used in the experiment is a 0.3 mW diode module with a wavelength of 633 nm. The two LCOS SLM devices have a resolution of 1920 × 1080 with a pixel pitch of 5.3 µm and a grey level scale of 0–255. A linear polarizer (LPVISC050) with a diameter of 12.5 mm is used to create the input light with a linear polarization angle of 0° for the system. A polymer zero-order half-wave plate (WPH05ME) with a diameter of 0.5 inches is used to rotate the polarization angles of two separate light components. Two beam splitters (BS004) are used, comprising the non-polarizing beam splitter cube with an energy split ratio of 50:50. A beam splitter is used to splitting the input linearly polarized beam into two with the same intensity. Another beam splitter is used to combine two orthogonally polarization beams into one beam with a final linear polarization state. The beam combiner may be replaced by other polarizing optical components, such as a Wollaston prism and Calcite beam displacer, to realize the recombining of two light beams. The FFT lens (LB1374) is an N-BK7 Bi-Convex Lens with a diameter of 50 mm and a focal length of 150 mm. The CMOS sensor has a pixel size of 5.3 µm and a resolution of 4928 × 3624.

### 3.2. The Superposition Process of Two Separate Vector Beams

The final stage of the phase and polarization modulation method is to combine two modulated holographic vector beams into a single vector beam. Since the horizontal and vertical components of the output holographic vector beam directly come from two basis vector beams, the accuracy of the beam combination can determine the quality of the output light field.

The combination of two light beams with vertical and horizontal polarization has been studied due to the huge demands in optical fiber applications. Therefore, a commercial fiber-based polarization beam combiner has been well developed. In this device, two beams are coupled into an optical fiber and then combined inside a polarizing beam displacer (e.g., calcite prism). However, this method of combining light is not suitable for the generation of data writing light field. There is a major reason: coupling the holographic light field into a fiber. When we couple a light beam into a fiber, normally an objective lens is used to focus light into the area of the fiber core. The focusing of the light beam is a simple step. However, this step cannot be applied to the holographic vector beam. Due to the principle of the Fourier hologram, the image of diffraction locates at an infinite distance away from the hologram. Thus, in the holographic imaging process, a converging lens will be used to reconstruct the image at its focal plane. This means that when we try to couple two basis holographic beams into fibers, the lens used for coupling will also image the whole holographic pattern. Therefore, the fiber-based polarization combiner cannot be utilized in the proposed phase and polarization method.

Considering the properties of the holographic light field, a method that allows the combination of light beams with holographic pattern information is proposed. This method is based on the superposition of two coherent laser beams. As shown in Figure 9, with this method, two polarized collimated beams are aligned and then spatially superposed into a single beam with a combined polarization state and amplitude by using a beam splitter. The superposition of two beams in free space does not require any focusing; therefore, the holographic information in the light field can be preserved in the combining process.

When this method is applied in a practical optical setup, the alignment of two input vector beams is critical for the combining process. First, to ensure the two beams being parallel, the input direction of the two beams needs to be normal to the surface of the cubic beam splitter. This is achieved by adjusting the angles of the LCOS SLMs surface, mirror, and beam splitter. The two parallel input beams will then be spatially overlapped at the interface inside the beam splitter by adjusting the position of a mirror in Figure 9.

To check the quality of beams combining, an analyzer at 45° is placed in front of the CMOS to examine the polarization state of the output beam. When two beams with the same amplitude and a phase shift are at 2nπ or nπ, the polarization angle of the superposition beam will be 45° or −45°. Therefore, a dark or bright beam pattern directly after the analyzer can indicate the final polarization state. Here, the experimental results of the two main steps of the beam combining process will be shown and discussed.

The first step is the parallelization of the two beams. In the combing process, the two orthogonal vector beams show similar behavior of light interference. When two beams have alignment error, as can be seen in Figure 10, there will be a distance between two beams on the interface of the beam splitter. This means that the phase difference of two beams at the different positions on the image plane will not remain 0 or 2π due to the difference in their optical path length. This is similar to when two coherent laser beams are generated from two slits and form-phase a different pattern on the image plane, the location constructive (light) and destructive (dark) interference pattern corresponds to the phase difference distribution.

According to the interference of two light beams with the same polarization, the phase difference of two beams at the center of dark and light stripes is 2nπ or nπ. Maintaining the same spatial phase difference distribution of the interference of light, when two beams have orthogonal polarization, the polarization state at the same place will become 45° or −45°. Therefore, as shown in Figure 11a–c, when two input beams are not spatially parallel, the dark and light patterns can be observed by using an analyzer at 45°.

An extending experiment shows that the interference pattern only appears at the superposition area of two vector beams with different beam shapes. Figure 11d,e shows the experimental results. This indicates that the superposition of two vector beams requires both spatial alignment and complete overlapping.

Moreover, Figure 12 shows that with an increase in the degree of parallelization, the width of dark and bright stripes increased and the number of stripes in the overlap of two beams decreased. The width *W* of stripes can be defined as:(9)$$W=\frac{d\lambda }{a}$$
where λ is the wavelength of light, *a* indicates the length of the gap between two beams on the interface inside beam splitter, and *d* is the distance between the image plane and the superposition point of two beams. When the distance *a* is close to 0, the width of stripes will increase to infinite. This means that the polarization state in the overlapped area can become uniform.

When the beam alignment is well performed but two beams are not completely overlapped, the overlapping area of two vector beams will have a uniform polarization state. This means that no interference pattern can be observed under an analyzer. Thus, we can use the disappearance of interference patterns as a sign of complete alignment. Figure 13 and Figure 14 show the beams’ superposition results after high-accuracy alignment. In the experiment, two basis vector beams are aligned in parallel for 15 m to ensure a high beam combining quality. The images are taken under the analyzer that is placed at 45° and −45°. We can see that the overlapped area of the two beams does not have any interference patterns. This means that the light field in the overlapped area has a single linear polarization state.

As shown in Figure 14, when two orthogonal vector beams are successfully combined into one single vector beam, the amplitude and polarization angle of the output beam can be determined by two factors, respectively: the absolute amplitude of two input beams and their amplitude ratio. In this experiment, two input vector beams have the same amplitude and the overlapped area after beam combining has a polarization angle of 45°. As shown in Figure 15, when the analyzer is set at 45°, the intensity of the superposition area is about $2\sqrt{2}$ times compared with the intensity of input beams (not overlapped part). This result is further proof that the superposition of two vector beams in the overlapped area is achieved. By completely overlapping the two vector beams, the final superposition process of the two vectors can then be completed.

In summary, this experimental result indicates that two vector beams can be directly combined into one single beam in free space using a beam combiner. In addition, a precise alignment calibration in the order of a wavelength is necessary. This means that two vector beams with independent amplitude information and the same holographic information can be combined into one single light field. Therefore, the polarization angle of the output light field can be actively modulated without the loss of holographic information. This ensures the feasibility of the phase and polarization method in use for data writing light field generation.

### 3.3. Experimental Results of Holographic Vectorial Image

In this section, the effectiveness of the phase and polarization modulation method is experimentally verified. To generate a data writing field, we need a data pattern image with a certain polarization state as the output light field. Therefore, the holographic information on both LCOS SLMs needs to be identical, while the pre-calculated amplitude information needs to be applied to the CGH. The two different sets of amplitude information are encoded by using two-phase depths’ values on the LCOS SLMs according to the lookup table created in Section 2.3. Figure 16 shows the results at different polarization states. When the specific phase depth on both LCOS SLMs is determined, the target polarization state of the output holographic light field is formed.

First, a pair of phase depth combinations on the two LCOS SLMs is applied to generate a target image with a target polarization angle of 30°. The target image used in the experiment is a double hexagon pattern. As can be seen from Figure 16, the brightest image and the light cancellation are observed when the analyzer’s polarization angle is at 30° and 60°, respectively. We also find that the average image intensity when the analyzer is at 30° has a clear trigonometric relationship with the average image intensity recorded when the analyzer is at 0° and 90°. This means that the final linear polarization state is the superposition of two basis vector components. Therefore, we show the successful control of the target image and the target polarization state using the proposed method.

Moreover, we can use another pair of phase depth combinations on the two LCOS SLMs to generate the image with a linear polarization angle at 45°. Similarly, we can see that the brightest image is observed when the analyzer’s polarization angle is at the target polarization angle of 45°, while the light cancellation is recorded at −45°. Figure 17 shows the different intensities of target images when the analyzer is placed at different angles. We also find that the average image intensity when the analyzer is at 45° has a clear trigonometric relationship with the average image intensity recorded when the analyzer is at 0° and 90°.

At last, we generate the same pattern with the linear polarization angle at 60° to further prove the effectiveness of our method. The brightest image and the darkest image are observed at the designed polarization angle and its crossed angle in Figure 18. We also find that the average image intensity when the analyzer is at 60° has a clear trigonometric relationship with the average image intensity recorded when the analyzer is at 0° and 90°. The results from Figure 16, Figure 17 and Figure 18 confirm that each holographic pattern has an expected polarization state.

These results show that the proposed phase and polarization method can simultaneously achieve phase modulation (holographic image pattern) and polarization modulation (the angle of polarization state). Moreover, with this method, both phase and polarization modulations are dynamically controlled by a CGH on LCOS SLM devices. No external complex or active optical elements are required. This advantage can play an important role in the application of data writing beam generation. The proposed method can also be applied in the field of holographic displays, such as holobricks [44].

## 4. Discussion

After verifying the effectiveness of the proposed method of phase and polarization modulation, the comparison and discussion of the proposed method and existing method for the application of optical data storage in a glass follows.

First, we will summarize the demand for the generation of writing beams for optical data storage in a glass. The polarization angle of a voxel is one of the parameters used in data bits encoding. Thus, a data page in the glass block consists of voxels with different polarization angles, and the voxels with the same polarization angle can have an arbitrary location distribution according to the data encoding. Therefore, the writing beam of voxels needs to be able to produce a random multi-beam pattern, at the same time, while the multi-beam light field needs to have the target linear polarization state. Both phase modulation and polarization modulation of light are required at the same time for the generation of the data writing beam. We also compared the proposed method (two beams combination) and a current typical method (i.e., LCOS+HWP) in the aspects of modulation scheme and application performance.

The first aspect is the phase modulation. As for the current methods, the target multi-beam pattern for voxel writing is generated by holographic imaging using phase-only LCOS SLMs. In the proposed method, the target image of the data pattern is also provided by a computer-generated hologram using LCOS devices. Both methods utilize phase-only modulation to encode the target image information.

The second aspect is the polarization modulation. To modulate the polarization state of light, the half-wave plate is used to rotate the original linear polarization state into the target state in the existing methods. For linearly polarized light, the effect of the half-wave plate (HWP) is to rotate the polarization vector through an angle that is twice the angle between the incident polarization vector and the half-wave plate’s optic axis (usually the fast axis). Thus, to acquire one target polarization angle, the optic axis of HWP needs to be at a specific angle. Another improved method using HWP is to fabricate an HWP matrix. This method is the one chosen for comparison with the proposed method. The HWP matrix is a combination of four HWP with the different optic axis that can provide target polarization angles of 0°, 45°, 90°, and 135°. In this method, the rotation of a single HWP is replaced by moving the glass sample underneath the HWP matrix film. However, the principle of polarization modulation is still based on the use of HWP. In the proposed method, in the phase-only modulation by computer-generated hologram, not only the target data image information has been encoded onto the input beam but the amplitude information has also been encoded. Two input beams with orthogonal polarization states are parallelly modulated by two LCOS SLMs. After the reconstruction of two independent holographic images with predetermined intensity, the two light fields are spatially combined into one final output. The control of the angle of linear polarization is achieved by the direct combing of two basis beams with a predetermined amplitude ratio.

The third aspect is multiple polarization states writing. In the writing process of a data page in glass, more than one target polarization angle is required, and one of the current methods utilizes a rotatable HWP. The rotatable HWP provides the target polarization state by rotating to the correct position. It needs to synchronize a motor-driven HWP rotator to the LCOS SLM and the femtosecond laser. Thus, the complicity of the writing beam generating system is largely increased. Moreover, the mechanical delay of the HWP rotator will accumulate at each time of polarization change, resulting in a large reduction of the data writing speed. The improved method using the HWP matrix can avoid HWP rotation. However, it still requires the synchronization of the CGH refresh on LCOS SLMs and the glass substrate movement. When the current polarization of the writing beam needs to be changed into a new one, the glass substrate will be moved to the correct position under the HWP matrix so that the polarization state of writing beams can be altered into the target one. In the proposed method, because it uses the spatial combining of two basis beams to form any target polarization state, there is no need for any external polarization modulator (HWP matrix system). This means that the complicity and difficulty in synchronization are largely reduced. The proposed method also does not require any glass substrate movement in the writing of each data page at a certain depth. The switching of different writing beams’ polarization states is achieved by loading new CGHs on LCOS SLMs.

The fourth aspect is the delay, which is also an important factor. The current method using the HWP rotator has a large delay because the rotation takes a relatively long time. To provide any possible polarization angle from a fixed incident polarization state, the required maximum rotation range between switching can be 87.5° (the resolution of voxel polarization forming is 5° [45]). The current method using the HWP matrix has less delay compared with the rotator; however, it still requires many glass sample movements for the writing of one data page. Due to the use of a motor-driven platform, the mechanical delay caused by the acceleration and deceleration processes takes up a major part of the current writing process time duration. The proposed method can provide motion-free writing for each data page. Because the only active optical elements used in this method for polarization control are phase-only LCOS SLMs, the switching time between two polarization states in writing is only determined by the refresh rate of CGHs on LCOS devices (60 Hz~100 Hz for nematic LCOS devices).

The fifth aspect is the extensibility in the available polarization state. The current method using the HWP matrix can only have a limited number of available polarization states for data writing. This is because the number of HWPs with the different fast axis on one HWP matrix is fixed after fabrication. When more polarization states are required in data writing, the only solution is to fabricate a new HWP matrix with more sub-sections. Apart from the workload and difficulty in fabrication, the new HWP matrix can have a larger area due to the increase of HWPs. A larger area means a longer range of glass substrate movement in data writing, causing a further increase in delay. The proposed method can simultaneously control the holographic target image and its polarization state by two parallel phase-only modulations. After the alignment in the two beams’ combining process is finished, the optical setup will become a passive component in the whole system. Any change in the writing beams can be dynamically accomplished by altering the CGHs on LCOS SLMs.

## 5. Conclusions

A phase and polarization modulation method based on LCOS SLMs capable of the simultaneous implementation of dynamically generating holographic patterns and arbitrary linear polarization states is proposed, with the effectiveness of the method experimentally verified. In the proposed method, we can independently use phase-only modulation to encode the holographic target image information and amplitude information onto two input beams. After obtaining the two modulated light beams with target pattern and target amplitude ratio, the spatial superposition of two beams can provide one output light field with the target polarization state. Therefore, the polarization control in the proposed method is achieved by the manipulation of the amplitude ratio between vertically and horizontally polarized beams and the combination of these two basis beams. The superposition of two beams in the proposed method requires a precise alignment process to ensure high accuracy and quality output image and polarization. The spatial overlap of the two beams needs to be in the order of wavelength. Although there are methods that can generate a similar number of polarization states, they require an external polarization modulator after data pattern generation. To alter the holographic target image and the target polarization state of data writing beams, we only need to load new CGHs on LCOS SLMs. Therefore, the proposed method can largely increase the data writing speed and make this data storage technology more feasible for the future cloud. It is noted that the combination of two polarized beams is highly sensitive to environmental disturbances. The experimental setup should be constructed in a good vibration-isolation environment. In addition, the collimation property of two beams has a high requirement. Thus, the two-beam collimation calibration is also necessary using a longer optical path. In the next step, in the future, this method will be utilized in the actual data writing task. When a glass sample is utilized in the writing optical path, a challenge is that the distortion of the focused points in the glass happens because the glass media is heterogeneous. An adaptive optics approach will be a good solution for this issue in the future.

## Figures and Tables

**Figure 1 sensors-22-09010-f001:**
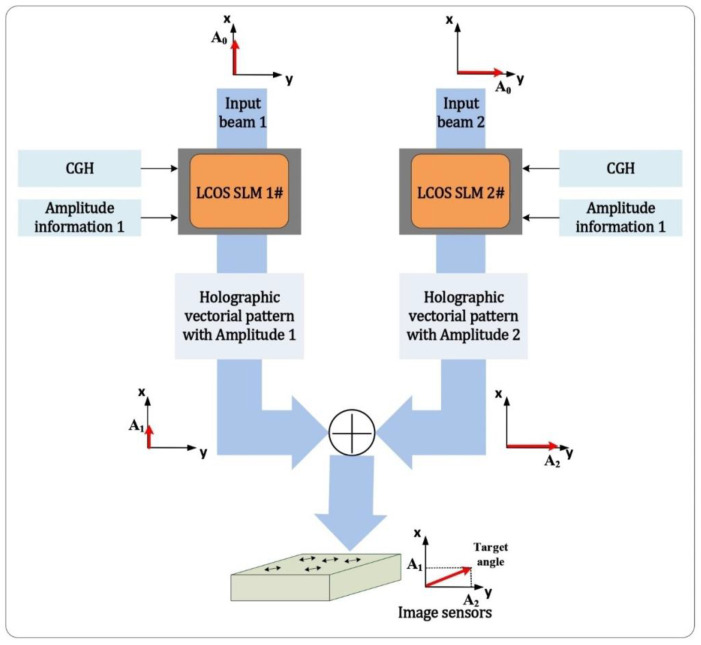
Schematic of the proposed method using two-beam parallel coding in combination. The information of the target image and amplitude are independently encoded onto the input beams to form the target polarization state.

**Figure 2 sensors-22-09010-f002:**
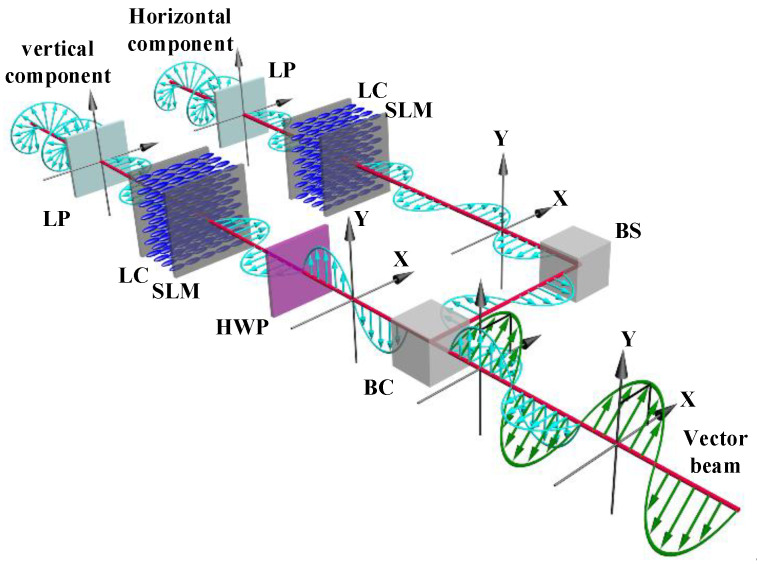
The principle of the proposed method using a tunable optical vector, where LP is a linear polarizer, LC is liquid crystals, HWP is a half-wave plate, BS is a beam splitter, and BC is a beam combiner.

**Figure 3 sensors-22-09010-f003:**
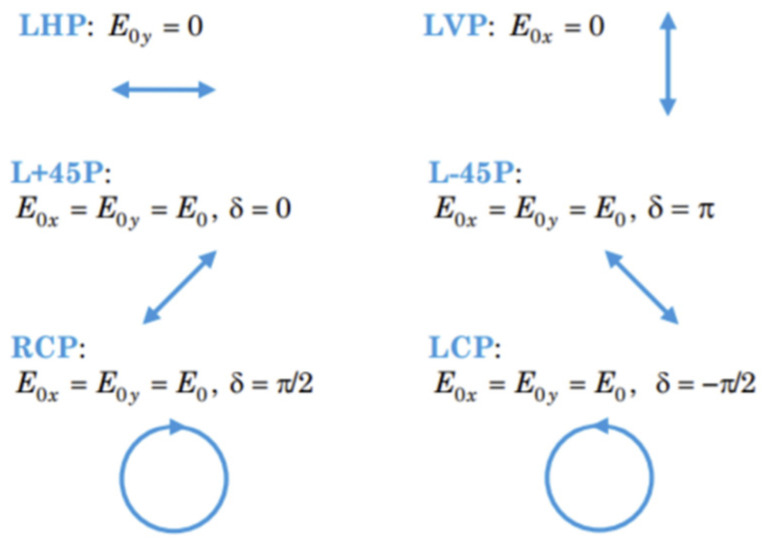
An example of the different polarization states determined by the amplitude ratio and phase shift between two polarization components.

**Figure 4 sensors-22-09010-f004:**
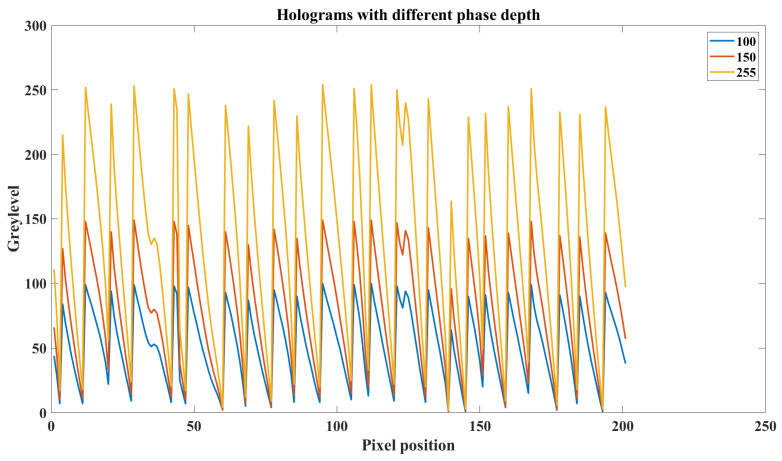
The partial phase profile of one hologram under three different phase depths.

**Figure 5 sensors-22-09010-f005:**
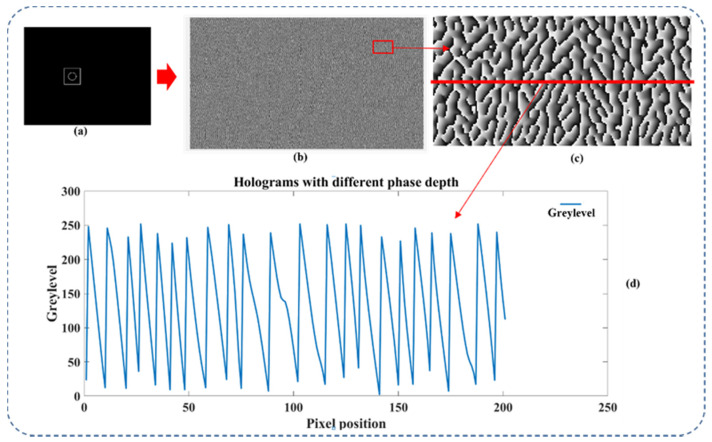
The partial phase profile of the hologram with three different grey levels displayed on the active area of an LCOS SLM: a target image (**a**), the corresponding CGH hologram (**b**), a selected area of the hologram (**c**), and the grey levels (**d**).

**Figure 6 sensors-22-09010-f006:**
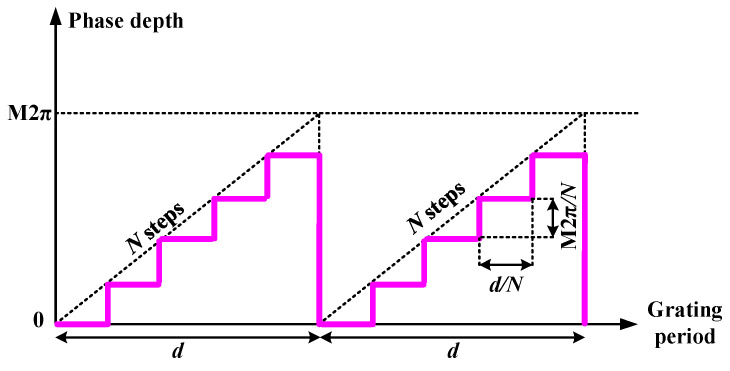
Phase profile of a blazed phase stepped grating on an LCOS SLM with period d, maximum phase depth M2π, and N steps.

**Figure 7 sensors-22-09010-f007:**
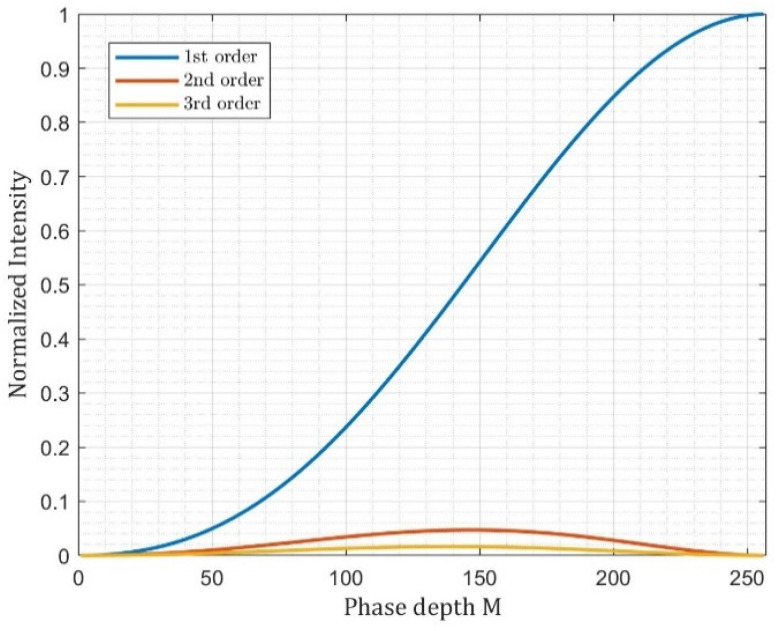
The intensity of different diffraction orders (μ=1, 2, 3) versus the phase depth *M* of a continuous phase grating (N→∞).

**Figure 8 sensors-22-09010-f008:**
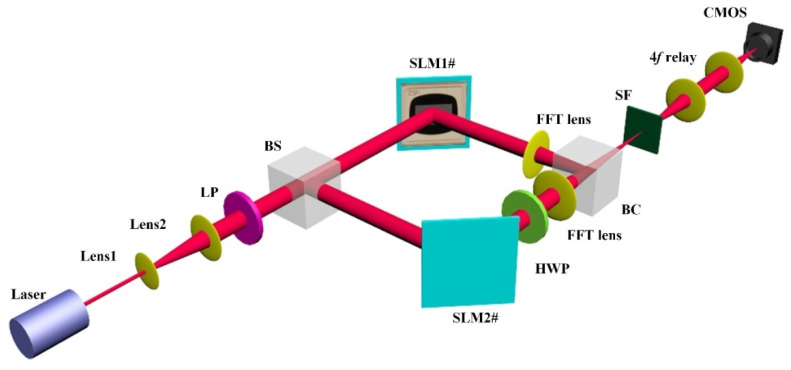
Schematic of the experimental setup, where LP is a linear polarizer, HWP is a half-wave plate, BS is a beam splitter, BC is a beam combiner, and SF is a spatial filter.

**Figure 9 sensors-22-09010-f009:**
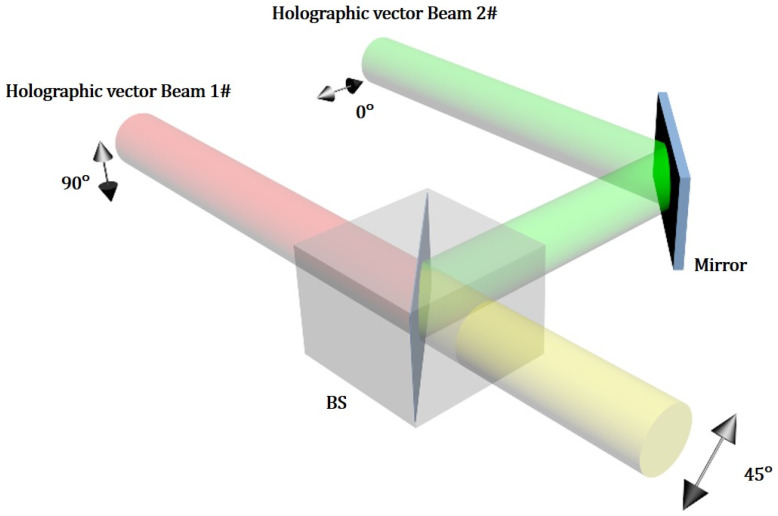
Schematic of the proposed method using spatial superposition of two holographic beams with orthogonal polarization states. The output of combining beams will have a 45° polarization angle when the amplitude ratio of two input beams is 1:1.

**Figure 10 sensors-22-09010-f010:**
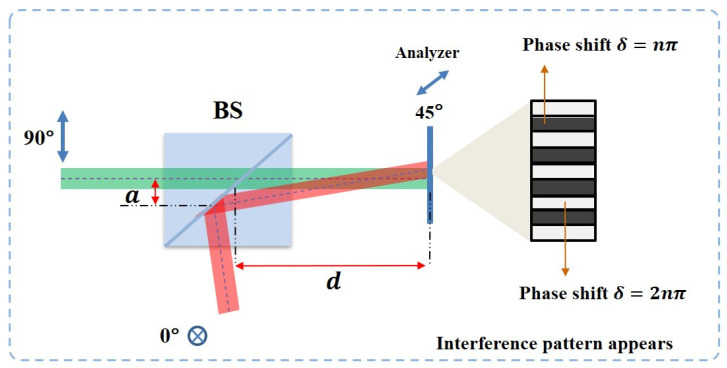
When two basis vector beams have an alignment error, two orthogonal vector beams will have a small angle at the image reconstruction plane. Due to the high coherence of two beams, they will spatially interfere. Therefore, the interference fringes can be observed under an analyzer.

**Figure 11 sensors-22-09010-f011:**
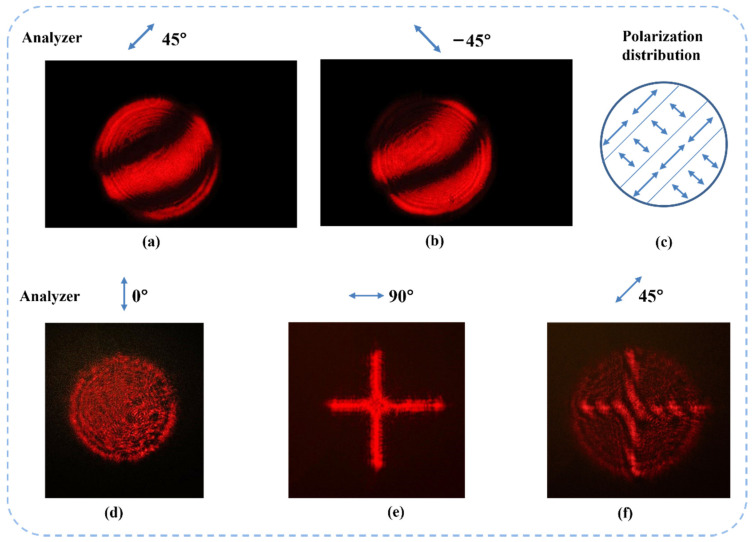
Interference patterns of two orthogonal vector beams with the same shape are observed when the analyzer is placed at 45° (**a**) and −45° (**b**). The combined beam includes two original polarization states (**c**). (**d**,**e**) show the superposition of two orthogonal vector beams with different shapes (circle and cross) that is observed under different polarization analyzer angles (blue double-end arrow). Interference patterns are observed when the analyzer is placed at 45° (**f**).

**Figure 12 sensors-22-09010-f012:**
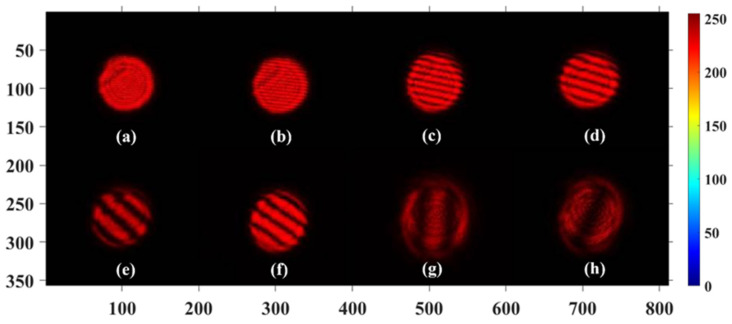
Experimental results (**a**–**h**) of unparallel two vector beams with the same shape combining in free space. The width-of-stripes pattern varies with the degree of parallelization of the two beams.

**Figure 13 sensors-22-09010-f013:**
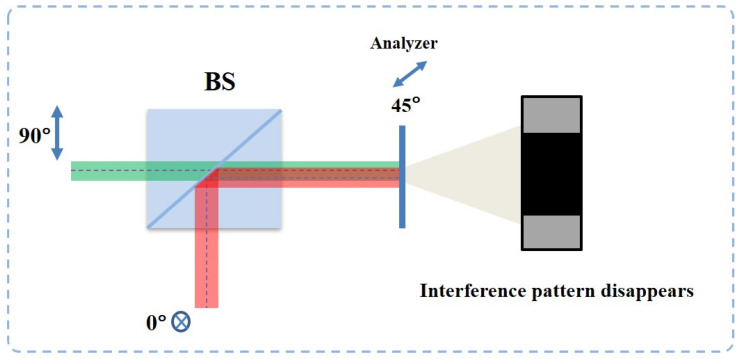
The two orthogonal vector beams are completely parallel after the high-accuracy aligning process and the interference pattern disappears due to the combination of two beams.

**Figure 14 sensors-22-09010-f014:**
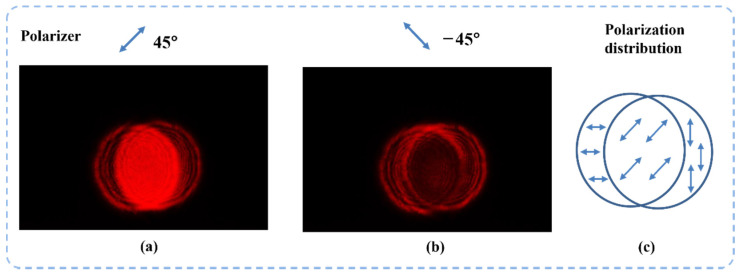
The two orthogonal vector beams are completely aligned. The superposition area of two beams has a single polarization state. (**a**) is the result when the polarizer is placed at 45°, (**b**) is the result when the polarizer is placed at –45°, and (**c**) is the polarization state of the combined beams.

**Figure 15 sensors-22-09010-f015:**
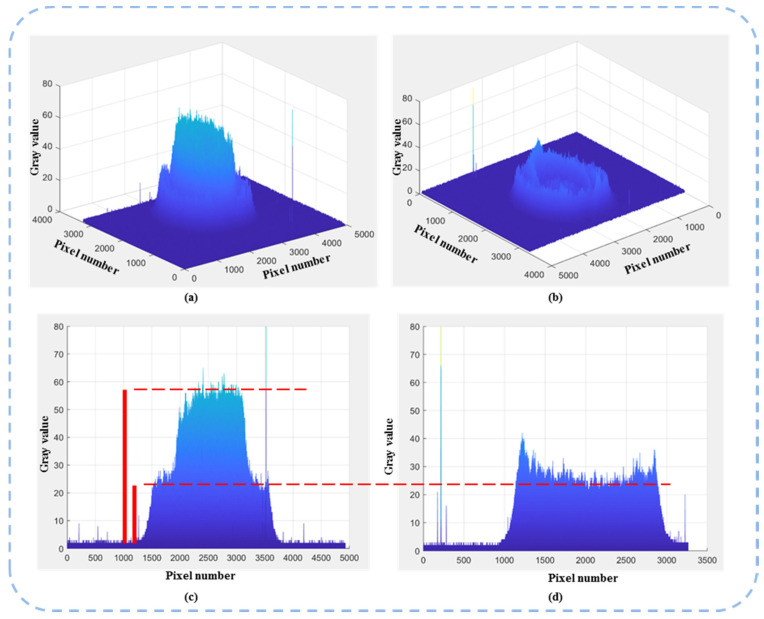
The 3D plots of the intensity of images provide a comparison of the overlapped area and area of input beams under an analyzer at 45°. (**a**,**b**) and (**c**,**d**) show the corresponding 3D intensity and 2D intensity, respectively.

**Figure 16 sensors-22-09010-f016:**
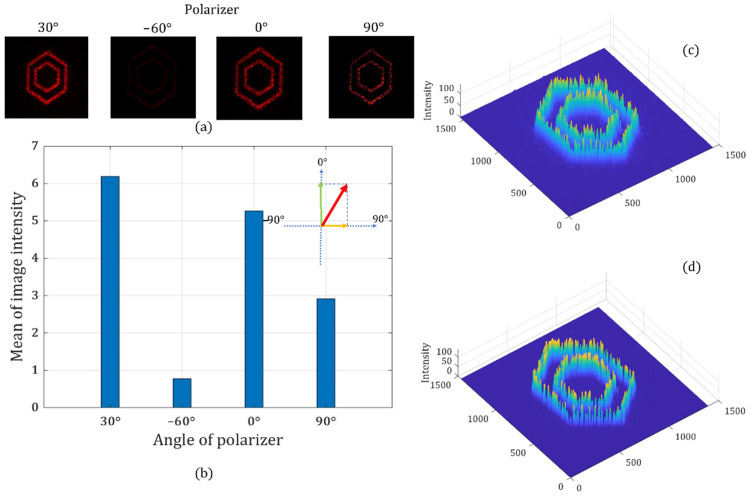
Holographic images with a linear polarization state of 30°: recorded holographic images when the analyzer is set at different angles (**a**), the average intensity of four holographic images (**b**), the 3D intensity distribution at 30° (**c**), and the 3D intensity distribution at 0° (**d**).

**Figure 17 sensors-22-09010-f017:**
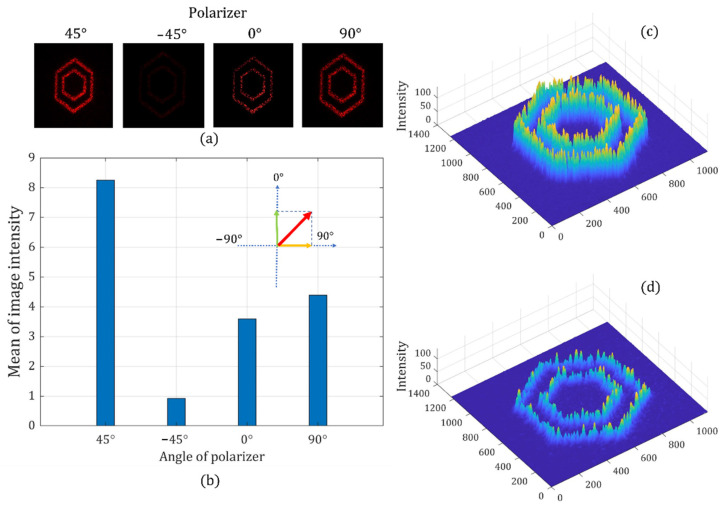
Holographic images with a linear polarization state of 45°: recorded holographic images when the analyzer is set at different angles (**a**), the average intensity of four holographic images (**b**), the 3D intensity distribution at 45° (**c**), and the 3D intensity distribution at 0° (**d**).

**Figure 18 sensors-22-09010-f018:**
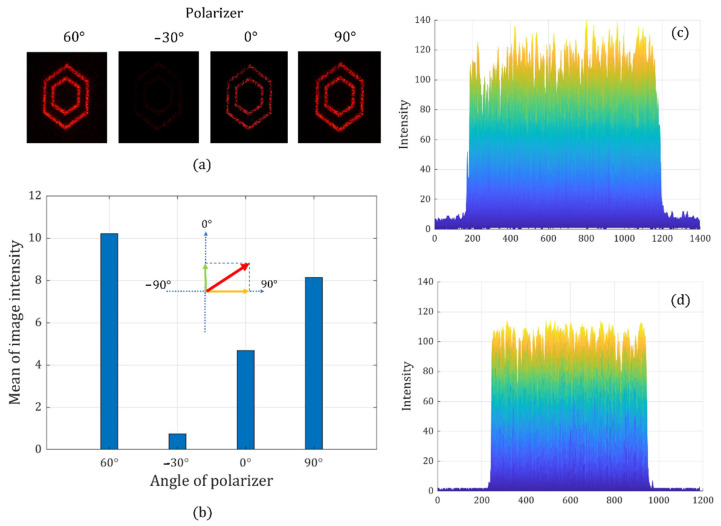
Holographic images with a linear polarization state of 60°: recorded holographic images when the analyzer is set at different angles (**a**), the average intensity of four holographic images (**b**), the 3D intensity distribution at 60° (**c**), and the 3D intensity distribution at 90° (**d**).

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
