# Peer review of "Dynamic Phase and Polarization Modulation Using Two-Beam Parallel Coding for Optical Storage in Transparent Materials"

_sensors, 2022, doi:10.3390/s22229010_

Round 1

Reviewer 1 Report

In the manuscript titled “Dynamic phase and polarization modulation using two-beam parallel coding for optical storage in transparent materials,” the authors present a modulation technique to control linear polarization states and holographic patterns.

Although the research topic that is addressed is interesting, I found major flaws in the manuscript, so I do not recommend publishing this work. I recommend resubmitting it after major revisions

Comment 1: In reading the manuscript, page. 3, line 146, seems a “3” is missing. Maybe should be written, “In section 3…”.

Comment 2: I suggest merging figures 11 and 12, identifying each part with letters or numbers to facilitate reading.

Comment 3: Figure 13 needs to be improved. The degree of parallelization of the two beams in each intensity image of the figure must be indicated. I suggest identifying with letters or numbers, indicating the scale (with a bar for example) and resolution of each image.

Comment 4: Figure 16 needs to be improved. The graphs must have a name, the axes also require a name and units. In addition, it facilitates reading to identify each image of the figure with a letter or number. I also suggest more details in the figure caption.

Comment 5: In reading the manuscript, page. 16, lines 521-522, it looks like a table is being referenced, it's not clear, and I didn't find any table in section 2.3 (in fact the table on page 18 is called “table 1”).

Comment 6: In figure 19, why are graphs (c) and (d) not in isometric perspective as in figures 17 and 18?

Comment 7: Table 1 can be greatly improved. Adding more methods (named in the introduction) would allow the reader to clarify the differences. Also, it seems to me that the table requires quantifiable quantities. In fact, some data named in the text can be added.

Question 1: 

The authors highlight the advantages of the proposed method throughout the manuscript. In my opinion, it is also necessary to clarify the limitations of the technique. The parallelization stage of the two beams seems crucial for the correct functioning of the technique. How sensitive is the experimental setup to environmental disturbances? This needs to be addressed in more detail considering that many holography methods that require a reference beam present difficulties in their implementation.

According to the manuscript, the use of the technique in data writing was not verified; what would be the challenges and the next steps for its implementation in that application (in which a lot of emphasis is made)? This, in my opinion, should be clarified in the conclusions in greater detail.

Question 2: 

What is the resolution with which the technique can control the amplitude ratio? Considering that the polarization states depend on the efficiency of the SLM, how many different ones can be generated? Of those that can be generated, do they all have the same precision? if not, it is desirable to detail the error and adds a discussion of the different types of modulators, as their characteristics may differ.

Author Response

In the manuscript titled “Dynamic phase and polarization modulation using two-beam parallel coding for optical storage in transparent materials,” the authors present a modulation technique to control linear polarization states and holographic patterns.

Although the research topic that is addressed is interesting, I found major flaws in the manuscript, so I do not recommend publishing this work. I recommend resubmitting it after major revisions

Response: Thank you very much for your suggestions. We have revised this manuscript based on your suggestions.

Comment 1: In reading the manuscript, page. 3, line 146, seems a “3” is missing. Maybe should be written, “In section 3…”.

Response: Yes, we correct this typo.

Comment 2: I suggest merging figures 11 and 12, identifying each part with letters or numbers to facilitate reading.

Response: Yes, we merged figures 11 and 12 and use the letters to identify each part.

Comment 3: Figure 13 needs to be improved. The degree of parallelization of the two beams in each intensity image of the figure must be indicated. I suggest identifying with letters or numbers, indicating the scale (with a bar for example) and resolution of each image.

Response: Yes, we improved Figure 13 and indicated the parallelization of two beams. We also indicate the scale and resolution of each image.

Comment 4: Figure 16 needs to be improved. The graphs must have a name, the axes also require a name and units. In addition, it facilitates reading to identify each image of the figure with a letter or number. I also suggest more details in the figure caption.

Response: Yes, we improved Figure 16.We also added the names of the graphs and axes. The units of  the axes are “1”. We also added the letters for identifying each image.

Comment 5: In reading the manuscript, page. 16, lines 521-522, it looks like a table is being referenced, it's not clear, and I didn't find any table in section 2.3 (in fact the table on page 18 is called “table 1”).

Response: In section 2.3, a look-up table is a relationship between the phase depth and the intensity.

Comment 6: In figure 19, why are graphs (c) and (d) not in isometric perspective as in figures 17 and 18?

Response: In Figure 19, we would like to show the intensity value difference of two different polarization angles.

Comment 7: Table 1 can be greatly improved. Adding more methods (named in the introduction) would allow the reader to clarify the differences. Also, it seems to me that the table requires quantifiable quantities. In fact, some data named in the text can be added.

Response: Yes, actually, we have described the information of Table 1 in the following text. We removed Table 1.

Question 1: 

The authors highlight the advantages of the proposed method throughout the manuscript. In my opinion, it is also necessary to clarify the limitations of the technique. The parallelization stage of the two beams seems crucial for the correct functioning of the technique. How sensitive is the experimental setup to environmental disturbances? This needs to be addressed in more detail considering that many holography methods that require a reference beam present difficulties in their implementation.

According to the manuscript, the use of the technique in data writing was not verified; what would be the challenges and the next steps for its implementation in that application (in which a lot of emphasis is made)? This, in my opinion, should be clarified in the conclusions in greater detail.

Response: Yes, we clarified them in the conclusions.

“It is noted that the combination of two polarized beams is highly sensitive to environmental disturbances. The experimental setup should be constructed in a good vibration-isolation environment. In addition, the collimation property of two beams has a high requirement. Thus, the two-beam collimation calibration is also necessary using a longer optical path. In the next step, this method will be utilized in the actual data writing task in the future. When a glass sample is utilized in the writing optical path, a challenge is that the distortion of the focused points in the glass happens because the glass media is heterogeneous. An adaptive optics approach will be a good solution for this issue in the future.”

Question 2: 

What is the resolution with which the technique can control the amplitude ratio? Considering that the polarization states depend on the efficiency of the SLM, how many different ones can be generated? Of those that can be generated, do they all have the same precision? if not, it is desirable to detail the error and adds a discussion of the different types of modulators, as their characteristics may differ.

Response: The intensity of each diffraction order can be calculated from the Fourier series coefficients of the phase grating. It is a straightforward calculation to derive the Fourier coefficients of the stepped phase grating. Therefore, by analysing the Fourier coefficients of a grating, the relative efficiency of each order diffraction order can be acquired.

When we consider a case for a continuous phase grating where the step number is infinite, the resulting diffraction efficiency can be expressed as

When , the energy in first order diffraction is zero. When the phase depth changes in the range , the energy splits on the first order varies. When , the energy is entirely in the first order. The amplitude of first order diffraction of the blazed phase gratings has a direct positive correlation with the maximum greylevel used on the hologram.  

In practical experiments, the resolution of phase modulation depth is discrete 0 to 255 greylevels. This means that the amplitude of first of diffraction can have 256 different values between 0 to 1 (after normalized). As for the precision, as we can observe from the curve in the figure above, the intensity modulation using phase depth in very low depth (0 to 60) and very high depth (220 to 255) is very ineffective.

In the practical experiments, holograms of arbitrary target images will be used instead of a blazed grating. Therefore, a look-up for the phase depth and their corresponding first-order diffraction intensity of the LCOS SLM used in the research can be found. Due to the performance of LCOS SLM in practical will be affect many factors (liquid crystal cell uniformity, liquid crystal alignment, pixel pitch size etc.), a precise look-up of the first-order light intensity and the greylevel used on a LCOS SLM device is critical for the accurate polarization control of output beam.

Figure. the intensity of the first diffraction order at the different grey levels, the grey level controls the phase depth used in modulation.

To achieve accurate output polarization states, in practical, we will use the phase depth range that can give us relatively linear relation between amplitude and greylevels.

We believe use LCOS SLMs as the light modulator still the best choice for our objective. This is because the easy access of arbitrary image encoding (computer-generated holograms by using GS algorithm). Secondly, due to the optimization of phase flickers and liquid crystal driving method, the number of greylevels available from 0 to 2π can be increased to 512 or even 1024. This can enable the higher resolution of first order amplitude modulation, therefore, the higher resolution of output polarization state.

Reviewer 2 Report

In this paper, the authors suggest novel polarization modulation setup with the simultaneous generation of the arbitrary phase and polarization modulations for the future applications of laser beam generations. This method can provide a significant increase in information density stored in each voxel. Even though the working principle does not include novel physical insight, the proposed engineering method and results of study provides originality and detailed description with rigorous experimental verification. 

:The references cited in the paper are appropriate and the discussion on the literature (in the introduction) is solid and sufficient to provide understanding on the necessity and motif of this work. Especially, classification and analysis of polarization modulating methods in holographic recording are well written in depth.

I suggest two minor revisions.

First, please shorten the abstract.

Second, several figures should be more clarified. The plots in Figure 4 and 5 should be revised in style. The plots in the figures should be more clarified in their style. The plots in the figure 5 and 6 should be revised in temrs of font size and the author should not use image capturing. Moreover, the resolution of all figures could be improved.

Author Response

In this paper, the authors suggest novel polarization modulation setup with the simultaneous generation of the arbitrary phase and polarization modulations for the future applications of laser beam generations. This method can provide a significant increase in information density stored in each voxel. Even though the working principle does not include novel physical insight, the proposed engineering method and results of study provides originality and detailed description with rigorous experimental verification. The references cited in the paper are appropriate and the discussion on the literature (in the introduction) is solid and sufficient to provide understanding on the necessity and motif of this work. Especially, classification and analysis of polarization modulating methods in holographic recording are well written in depth.

Response: Thank you very much for your positive suggestions.

I suggest two minor revisions.

First, please shorten the abstract.

Response: Yes, we shorten the abstract.

“In this paper, we propose and experimentally demonstrate a parallel coding and two-beam combining approach for the simultaneous implementation of dynamically generating holographic patterns at their arbitrary linear polarization states. Two orthogonal input beams are parallelly and independently encoded with the same target image information but the different amplitude information by using two-phase computer-generated holograms (CGH) on two Liquid-Crystal-on-Silicon-Spatial-Light Modulators (LCOS SLMs). Then, two modulated beams are now considered as two polarization components and are spatially superposed to form the target polarization state. The final linear vector beam is created by the spatial superposition of the two base beams, capable of controlling the vector angle through the phase depth of the phase-only CGHs. Meanwhile, the combined holographic patterns can be freely encoded by the holograms of two vector components. Thus, this allows us to tailor the optical fields endowed with arbitrary holographic patterns and the linear polarization states at the same time. This method provides a more promising approach for laser data writing generation systems in the next-generation optical data storage technology in transparent materials.”

Second, several figures should be more clarified. The plots in Figure 4 and 5 should be revised in style. The plots in the figures should be more clarified in their style. The plots in the figure 5 and 6 should be revised in temrs of font size and the author should not use image capturing. Moreover, the resolution of all figures could be improved.

Response: Yes, we improved Figure 4 and Figure 5.  We also improved Figure 5 and Figure 6.

Round 2

Reviewer 1 Report

The authors answered my questions.

Minor comments:

(1) In figure 5 (d), the scale is at 255, however, the blue line says 100. Needs clarification.

(2) In figure 12, the color bar does not match the images. It needs to be clarified.

Author Response

Minor comments:
(1) In figure 5 (d), the scale is at 255, however, the blue line says 100. Needs clarification.
(2) In figure 12, the color bar does not match the images. It needs to be clarified.
Response: Thank you very much for your suggestions. 
1. we revised the blue lines in figure 5d. 
2. we corrected the color bar in Figure 12.
